# An Adaptive Refinement Scheme for Depth Estimation Networks

**DOI:** 10.3390/s22249755

**Published:** 2022-12-13

**Authors:** Amin Alizadeh Naeini, Mohammad Moein Sheikholeslami, Gunho Sohn

**Affiliations:** Department of Earth and Space Science and Engineering, Lassonde School of Engineering, York University, 4700 Keele Street, Toronto, ON M3J1P3, Canada

**Keywords:** depth estimation, optimization, deep learning

## Abstract

Deep learning has proved to be a breakthrough in depth generation. However, the generalization ability of deep networks is still limited, and they cannot maintain a satisfactory performance on some inputs. By addressing a similar problem in the segmentation field, a feature backpropagating refinement scheme (f-BRS) has been proposed to refine predictions in the inference time. f-BRS adapts an intermediate activation function to each input by using user clicks as sparse labels. Given the similarity between user clicks and sparse depth maps, this paper aims to extend the application of f-BRS to depth prediction. Our experiments show that f-BRS, fused with a depth estimation baseline, is trapped in local optima, and fails to improve the network predictions. To resolve that, we propose a double-stage adaptive refinement scheme (DARS). In the first stage, a Delaunay-based correction module significantly improves the depth generated by a baseline network. In the second stage, a particle swarm optimizer (PSO) delineates the estimation through fine-tuning f-BRS parameters—that is, scales and biases. DARS is evaluated on an outdoor benchmark, KITTI, and an indoor benchmark, NYUv2, while for both, the network is pre-trained on KITTI. The proposed scheme was effective on both datasets.

## 1. Introduction

Dense depth maps play a crucial role in a variety of applications, such as simultaneous localization and mapping (SLAM) [1], visual odometry [2], and object detection [3]. With the advent of deep learning (DL) and its ever-growing success in most fields, DL methods have also been utilized for generating dense depth (DD) maps and have demonstrated a prominent improvement in this field.

Depending on the primary input, DL-based depth generation methods can be categorized into depth completion and estimation methods. Depth completion methods try to fill the gaps present in input sparse depth (SD) maps [4,5], whereas depth estimation ones attempt to estimate depth for each pixel of an input image [6,7,8,9,10]. Although the results provided by depth completion methods [4,11] are usually more accurate than those from depth estimation ones, they need to be supplied by a remarkably large number of DD maps as targets in the training stage. This is while collecting such data in a real-world application is an expensive and time-consuming task [8,12].

In parallel, the input to depth estimation methods includes no depth maps; however, supervised ones take either SD [13] or DD maps [14,15,16] as the target during training. Between these two supervised depth estimation approaches, using SD maps usually leads to less accurate results but is more viable than using DD ones. Because SD-based methods only need SD maps which can be provided using a LiDAR sensor and without any need for post-processing or labeling effort. Considering the above issues, supervised depth estimation methods which use sparse depth maps are preferred, especially for most real-world cases in which access to large-enough accurate DD maps is difficult or even impossible.

Similar to all DL methods, DL-based depth estimation methods, no matter supervised or unsupervised, suffer from the generalization problem. In other terms, DL models trained on an arbitrary dataset are not able to preserve their satisfying performance neither on unseen data samples nor even on hard seen ones. This problem is more severe in applications such as SLAM and autonomous vehicles, where the test environment is under a constant change. Hence, the high variety in input samples leads to accuracy degradation [17].

In the segmentation field, a similar problem, i.e., limited generalization ability, has been alleviated by the backpropagating refinement scheme (BRS) [18]. This method has been proposed to optimize an input to a segmentation network based on user clicks. The method performs this process via backpropagating through the whole network; thus, it suffers from the computational burden. To speed up the process, feature BRS (f-BRS) [19] has been proposed to only backpropagate through several last layers and optimizes the activation responses of an intermediate layer using some introduced parameters. Optimizing those parameters during inference can be viewed as an adaptive approach because an intermediate activation function and indeed the output are adapted to input data. In other words, f-BRS converts the baseline network to a functionally adaptive method, in which the shape of activation function changes in the inference time [20,21].

As user clicks and SD maps are both sparse labels, it seems that the application of the above-mentioned scheme, f-BRS [19], can be generalized to depth estimation. Accordingly, we present an inference-time functionally adaptive refinement scheme for depth estimation networks (see Figure 1). For this purpose, f-BRS [19] is used, which can be injected into any DL baseline and adapts the model to hard and unseen environments or different datasets. Nevertheless, f-BRS suffers from two fundamental issues which prevent it from being applicable in depth estimation. The first problem is associated with its nature which cannot manage highly inaccurate products (here, the depth predicted by the network), since f-BRS has been originally proposed for interactive segmentation [19], where is often no need for a considerable modification. To resolve this, a sliced Delaunay correction (SDC) module is designed to carry out a correction using SD maps and provide an appropriate initial value for the optimizer. The second problem is that f-BRS is trapped in local optima due to its local optimizer. To address this, f-BRS is equipped with particle swarm optimization (PSO) as a global optimizer [22]. For simplicity, this novel generalization of f-BRS, which is applicable to depth estimation, is named as the double-stage adaptive refinement scheme (DARS), where the first stage is carried out through SDC and the second stage is conducted by optimizing the f-BRS parameters, i.e., adapting the activation maps.

Overall, our contributions can be summarized as:A novel double-stage adaptive refinement scheme for monocular depth estimation networks. The proposed scheme needs neither offline data gathering nor offline training, because it uses available pre-trained weights.Introduction of functional adaptation schemes in the field of depth generation, for the first time. Using the proposed adaptive scheme, pre-trained networks can be straightforwardly used for unseen datasets through adjusting the shape of activation functions of an intermediate layer.A model-agnostic scheme which can be plugged into any baseline. In this paper, we selected Monodepth2 [23] as one of the most widely used baselines for depth estimation.

## 2. Related Work

Here, we initially provide an overview for unsupervised and supervised depth estimation methods. In the last part, a brief review of functionally adaptive networks is displayed.

### 2.1. Unsupervised Depth Estimation Methods

These methods use color consistency loss between stereo images [24], temporal ones [25], or a combination of both [23] to train a monocular depth estimation model. Many attempts have been made to rectify the self-supervision by new loss terms such as left–right consistency [26], temporal depth consistency [27], or cross-task consistency [28,29,30]. Of these improvements, Monodepth2 has attracted substantial attention because of the different sets of techniques it has used for modification [23]. To the best of our knowledge, methods in this category have been presented for either outdoor environments, such as the above ones, or indoor environments, as in [31]. Not being applicable for both indoor and outdoor datasets can be regarded as a drawback of these methods. Another problem of these methods is that they suffer from low accuracy.

### 2.2. Supervised Depth Estimation Methods

The inputs to these methods are only images and they use either DD maps or SD maps as targets. This group can be categorized into DD-based and SD-based methods. Of these two, DD-based ones need DD maps during their training. DD-based methods, such as Adabins [14] and BTS [15], learn based on the error between predicted depth maps and DD maps. The main disadvantage of these methods is that they need DD maps for training.

Unlike DD-based methods, SD-based ones use SD maps only. Training data are not an issue for these methods because current robots and mapping systems can capture both images and SD maps simultaneously. The distance between predictions and SD maps are used as loss functions [32,33,34]. These methods are also known as semi-supervised [35].

### 2.3. Functionally Adaptive Neural Networks

Neural networks are called adaptive when they can adapt themselves to unseen environments, i.e., new inputs [36,37]. There are different techniques for designing adaptive networks, among which weight modification and functional adaptation can be mentioned. The former optimizes the network weights for new inputs while the latter modifies the slope and shape of the activation functions usually through a relatively few number of additional parameters [36]. Functional adaptation can be categorized under activation response optimization methods [38,39,40,41], in which the aim is to update activation responses while the network weights are fixed. The reason behind keeping the network weights fixed is to preserve the semantics learned by the network during the training process. On the other hand, one or several activation responses are modified to optimize the performance on inevitable unseen objects and scenes so that the network maintains its proficient performance in constantly changing environments [19].

The adaptation process can happen in either the training stage [20] or the inference stage for some tasks, such as interactive segmentation or SLAM, where some ground truth (even though sparse) is available on the fly [37]. In addition, the networks can adapt to a sequence of images or a single image. In a single-image adaptation, the core merit optimizes the prediction for a specific image or even an object, and the adaptation is discarded for the next image [37]. Thus, single-image adaptation can be beneficial, especially when scenes are prone to varying significantly.

Inspired from the biological neurons, some investigations have been conducted on the adaptive activation functions such as PReLU, which shows that adaptation behaviour in such activation functions can improve the accuracy and generalization of neural networks [20]. In [19], some parameters are introduced to adapt the activation functions to user clicks during the inference of the interactive segmentation task. An adaptive instance normalization layer is proposed in [21], which enables the style transfer networks to adapt to arbitrary new styles, adding a negligible computational cost.

## 3. Theoretical Background

In this section, some theoretical background needed for understanding the proposed scheme is provided. Firstly, Delaunay-based interpolation is explained, which is used in the correction stage to densify sparse correction maps. Subsequently, the particle swarm optimization (PSO) algorithm is displayed as the optimizer utilized in the optimization stage of the proposed scheme.

### 3.1. Delaunay-Based Interpolation

The first step of the interpolation is to conduct triangulation. Considering that there are many different triangulations for a given point set, we should obtain an optimal triangulation method, avoiding poorly shaped triangles. The Delaunay triangulation method has proved to be the most robust and widely used triangulation approach. This method connects all the neighbouring points in a Voronoi diagram to obtain a triangulation [42].

To find the value of any new point by interpolation, its corresponding triangle in which it lies should be identified. Suppose P(x,y) is a new point that belongs to a triangle with vertices of $P_{1}\left(x_{1},y_{1}\right)$, $P_{2}\left(x_{2},y_{2}\right)$ and $P_{3}\left(x_{3},y_{3}\right)$ with the values of $z_{1}$, $z_{2}$ and $z_{3}$, respectively, to linearly interpolate the value *z* of *P*, we should fit a plane (Equation (Equation 1)) to the vertices $P_{1}$, $P_{2}$ and $P_{3}$.
(1)z=ax+by+c

By inserting the known points $\left(x_{1},y_{1},z_{1}\right)$, $\left(x_{2},y_{2},z_{2}\right)$ and $\left(x_{3},y_{3},z_{3}\right)$ in Equation (Equation 1) and solving a linear system of equations, the unknown coefficients (a,b,c) of the plane are estimated. Finally, applying Equation (Equation 1) and having (a,b,c), the value *z* for any arbitrary point P(x,y) is interpolated within the triangle (Figure 2).

### 3.2. PSO

PSO is a population-based stochastic optimization technique inspired by the social behavior of birds within a flock or fish schooling [22]. PSO has two main components which need to be specifically defined for each application. One component is the introduction of particles, and the other is an objective function for particle evaluation.

Each particle has the potential of solving the problem; this means they must contain all the arguments needed for the problem in question.

The velocity and position of each particle are calculated using Equation (Equation 2) and Equation (Equation 3), respectively, [22]. Optimum values of unknown parameters are iteratively updated using the position equation, which is itself dependent on the velocity.
(2)$$V_{i}\left(t+1\right)=wV_{i}\left(t\right)+c_{1}r_{1}\left(t\right)\left[{pbest}_{i}\left(t\right)-X_{i}\left(t\right)\right]+c_{2}r_{2}\left(t\right)\left[gbest_{i}\left(t\right)-X_{i}\left(t\right)\right]$$
(3)$$X_{i}\left(t+1\right)=X_{i}\left(t\right)+V_{i}\left(t+1\right)$$

In Equation (Equation 2), $V_{i}\left(t\right)$ is the velocity of a particle *i* at time *t*, and ${pbest}_{i}\left(t\right)$ and ${gbest}_{i}\left(t\right)$ are personal and global best positions found by the particle *i* and all the particles by the iteration *t*, respectively. The *w* parameter is an inertia weight scaling the previous time step velocity. Parameters $c_{1}$ and $c_{2}$ are two acceleration coefficients that scale the influence of ${pbest}_{i}\left(t\right)$ and ${gbest}_{i}\left(t\right)$, respectively. In addition, parameters $r_{1}$ and $r_{2}$ are random variables between 0 and 1 obtained from a uniform distribution. The next position of each particle $\left(X_{i}\left(t+1\right)\right)$ can be calculated using Equation (Equation 3).

## 4. Proposed Method

Supervised depth estimation methods suffer from the generalization problem. In other words, they usually need to be retrained for achieving a proficient performance on an unseen dataset. To alleviate this, a double-stage adaptive refinement scheme (DARS) is proposed to equip pre-trained depth estimation networks with inference-time optimization for improving the performance on both seen and unseen datasets. The proposed scheme (Figure 3) consists of several components including a deep baseline model, a correction module which applies the first stage of refinement, and an activation optimization as the second stage. The baseline model could be any supervised or unsupervised pre-trained depth estimation network. The predicted depth by the baseline is given to the correction module to provide the optimization module with a sufficiently accurate depth map. In the second stage, scale and bias parameters are applied on a set of intermediate feature maps in the baseline, and they are optimized by a PSO to improve the accuracy of the final depth. The tasks and details of each module, and the overall proposed scheme, are displayed below. In the following subsections, *s* and *d* superscripts, respectively, indicate that depth maps are sparse or dense.

### 4.1. Baseline

Given an input monocular RGB image $I\in R^{w\times h\times 3}$, we rely on a depth estimation network $F:I\mapsto D_{0}^{d}$ to provide us with an initial depth map $D_{0}^{d}\in R^{w\times h}$. The proposed scheme can utilize any monocular depth estimation network. In this study, Monodepth2 [23] has been selected as the baseline, as one of most widely used depth estimation networks. The baseline is pre-trained and the weights are kept fixed.

### 4.2. Correction

The depth map $D_{0}^{d}$ predicted by the baseline lacks sufficient accuracy, especially for an unseen input. Thus, $D_{0}^{d}$ is not a proper initial value for the optimization stage. As a solution, in the first stage of the proposed refinement scheme, a sliced Delaunay correction (SDC) $C:R^{w\times h}\mapsto R^{w\times h}$ is used to correct $D_{0}^{d}$, using the available sparse depth map $D^{s}$. In SDC, first a correction value $\delta d^{s}\in \Delta D^{s}$ for any available depth pixel $d^{s}\in D^{s}$ is calculated: (4)$$\delta d^{s}=d_{0}^{s}-d^{s}$$
where $d_{0}^{s}\in D_{0}^{d}$ are the pixels in $D_{0}^{d}$ corresponding to the ones in $D^{s}$. Then, the sparse correction map $\Delta D^{s}$ is divided into three overlapped slices (see Figure 4). Neighbouring pixels are intuitively assumed to share a similar error pattern, and slices can represent a simplistic segmentation based on the error pattern.

In each slice, a Delaunay-based interpolation (see Section 3.1) $J:R^{2}\mapsto R$ is utilized to estimate a dense correction map $\Delta D^{d}=J\left(\Delta D^{s}\right)$, given the sparse one $\Delta D^{s}$. For the pixels in overlapped areas (see Figure 4), the average of the values coming from two adjacent slices is considered as the final depth correction value. As a result of this stage, a corrected depth ${\hat{D}}^{d}=D_{0}^{d}+\Delta D^{d}$ is generated, yet with marginal errors.

Regarding the number of slices, three was selected as the optimal number of slices on both datasets based on our experiments. Lower numbers could not result in homogeneous areas, and hence, a remarkable correction performance. On the other hand, a larger number was not considered because the improvement in accuracy was negligible with respect to the computational overload.

### 4.3. Activation Optimization

Given the initial value from the first stage (correction), the core part of network adaptation is conducted in the second stage. The technique chosen for the network adaptation is to modify an intermediate set of activation outputs [36]. This is usually carried out by freezing the weights and optimizing some auxiliary parameters. This way, not only are the valuable learned semantics preserved, but also the network can adapt itself to inputs. Inspired from works such as f-BRS [19] in an interactive segmentation field, we apply channel-wise scale and bias parameters on intermediate features of the baseline network. The scales are initialized to ones and biases to zeros; they are then optimized based on a cost function. To describe the algorithm of the optimization module better, the overall scheme, i.e., from the baseline to optimization module, is explained, followed by some details about the optimizer.

#### 4.3.1. Overall Scheme

Given an input RGB image $I\in R^{w\times h\times 3}$, denote the intermediate feature set as $G\left(I\right)\in R^{m\times n\times c}$ where $G:R^{w\times h}\mapsto R^{m\times n\times c}$ is the network body and *m*, *n*, and *c* are, respectively, width, height, and number of channels. The auxiliary parameters, scales $S\in R^{c}$ and biases $B\in R^{c}$ are applied on G(I), and the depth $D_{0}^{d}=H\left(S⨂G\left(I\right)⨁B\right)$ is predicted, where $H:R^{m\times n\times c}\mapsto R^{w\times h}$ is the network head, and ⨂ and ⨁ represent channel-wise multiplication and addition. Afterwards, the correction module $C:R^{w\times h}\mapsto R^{w\times h}$ carries out the first refinement stage on $D_{0}^{d}$ and returns ${\hat{D}}^{d}$:(5)$${\hat{D}}^{d}=C\left(D_{0}^{d},D^{s}\right)$$

The auxiliary parameters $X\in R^{2c}$, i.e., channel-wise scales and biases, are learnable. Therefore, the following optimization problem can be formulated as:(6)$$L\left({\hat{D}}^{d}\left(I,X+\Delta X\right),D^{s}\right)\to \min_{\Delta X}.$$
where ΔX is the corrections applied to the parameters and L is the cost function given to the optimizer.

#### 4.3.2. Optimizer

The above optimization problem can be given to any type of optimizers. The default optimizer of f-BRS is limited-memory Broyden–Fletcher–Goldfarb–Shanno (L-BFGS) [43,44]. This optimizer, due to its local gradient-based nature, is trapped in local optima. To overcome this problem, L-BFGS is replaced with PSO [22]. PSO iteratively updates scale and bias parameters in each particle based on the below distance loss:(7)$$L=\sqrt{\frac{1}{T}\sum_{i=1}^{T}∥\log\left({\hat{d}}^{s}\right)-\log\left(d^{s}\right)∥},$$
where *T* is the total number of pixels with depth values in $D^{s}$. Figure 5 shows the algorithm flow of the PSO and its parameters in DARS.

## 5. Experiments

In this section, we first briefly describe the datasets used in the experiments. Secondly, the metrics are introduced, and after that, an ablation study is discussed to show the effectiveness of each module. Finally, the results by the proposed scheme are compared with those of the state of the art.

### 5.1. Datasets

Two datasets are used in the experiments, KITTI [45] and NYUv2 [46]. KITTI is a well-known outdoor dataset, on which the baseline is trained, while NYUv2 is an indoor benchmark dataset and the adaptation performance of the scheme is highlighted through testing on it.

#### 5.1.1. KITTI

The KITTI dataset [45] consists of stereo RGB images and corresponding SD and DD maps of 61 outdoor scenes acquired by 3D mobile laser scanners. The RGB images have a resolution of 1241 × 376 pixels, while the corresponding SD maps are of very low density with lots of missing data. The dataset is divided into 23,488 train and 697 test images, according to [47]. For testing, 652 images associated with DD maps are selected from the test split. Sample data have been shown in Figure 6 for the KITTI dataset.

#### 5.1.2. NYUv2

The NYUv2 dataset [46] contains 120,000 RGB and depth pairs of 640 × 480 pixels in size, acquired as video sequences using a Microsoft Kinect from 464 indoor scenes. The official train/test split contains 249 and 215 scenes, respectively. Given that NYUv2 does not contain any SD maps, SD maps with 80% sparsity have been randomly synthesized from DD maps for the experiments of the proposed method. Sample data for NYUv2 dataset, including the synthetic SD maps, are illustrated in Figure 6.

### 5.2. Assessment Criteria

Assessment criteria proposed by [47] include error and accuracy metrics. The error metrics are root mean square error (RMSE), logarithmic RMSE (${\mathrm{RMSE}}_{\log}$), absolute relative error (Abs Rel), and square relative error (Sq Rel), whereas the accuracy rate metrics contain $thr=1.25^{t}$ where t=1,2,3. These criteria are formulated as follows:(8)$$RMSE=\sqrt{\frac{1}{T}\sum_{i\in T}\left|\right|d_{i}-d_{i}^{gt}{\left|\right|}^{2}},$$
(9)$$RMSE_{log}=\sqrt{\frac{1}{T}\sum_{i\in T}||log\left(d_{i}\right)-log\left(d_{i}^{gt}\right){\left|\right|}^{2}},$$
(10)$$AbsRel=\frac{1}{T}\sum_{i\in T}\frac{d_{i}-d_{i}^{gt}}{d_{i}^{gt}},$$
(11)$$SqRel=\frac{1}{T}\sum_{i\in T}\frac{\left|\right|d_{i}-d_{i}^{gt}{\left|\right|}^{2}}{d_{i}^{gt}},and$$
(12)$$accuracies=\%\;\mathrm{of}\;d_{i}\;\mathrm{subject}\;\mathrm{to}\;max\left(\frac{d_{i}}{d_{i}^{gt}},\frac{d_{i}^{gt}}{d_{i}}\right)=\delta <thr$$
where $d_{i}$ and $d_{i}^{gt}$ are the predicted and target (ground truth) depth, respectively, at the pixel indexed by *i*, and *T* is the total number of pixels in all the evaluated images.

### 5.3. Network Architecture

As the proposed scheme is by design model agnostic, the network architecture is not the focus of this study. Thus, we used the standard monocular version of the Monodepth2 [23] model with the input size of 640×192×3.

### 5.4. Implementation Details

We have used monocular Monodepth2 pre-trained on KITTI as our baseline. The input images were resampled to 640×192 and then were fed to the network. The weights were fixed and the network was run in inference mode. In SDC, the number of slices were three and the overlap between slices was set to 50%. Moreover, PSO paramters, i.e., $c_{1}$, $c_{2}$, number of particles, and number of iterations were, respectively, set to 0.5, 0.3, 10, and 30 in all the experiments. Furthermore, all the implementations were conducted in PyTorch [48].

### 5.5. Ablation Studies

This ablation study aims to prove the effectiveness of different stages and modules in the proposed scheme. To do this, starting from the baseline, we have enabled the correction and optimization modules in several steps (see Table 1). First of all, the result of Monodepth2 [23] with median scaling is discussed for comparison, while the version without any kind of post-processing (Monodepth2*) is also reported as our baseline. It means that the baseline results are without median scaling by target DD maps. As a result, they suffer from scale ambiguity and low accuracy. In addition, DC is introduced to show the efficacy of slicing in our proposed SDC as the correction module. The difference between SDC and DC is that, in the latter, Delaunay interpolation and correction are carried out on the entire depth maps instead of separately on each slice. For the sake of brevity, these two methods have just been surveyed for KITTI.

From Table 1, the worst results on KITTI in terms of all the metrics was recorded by the baseline (Monodepth2*), which was expected because of scale ambiguity. Using DC as the correction module improved the results by 13% in terms of RMSE, while SDC showed a significantly higher improvement over the baseline by 91%. This not only proves the contribution of the correction module but also indicates the effectiveness of the slicing process in SDC. Furthermore, SDC, without any use of target DD maps, yielded over 57% improvement with respect to Monodepth2, which means that SDC not only addresses the scale ambiguity problem but also corrects the given depth map significantly. Moreover, this observation supports the assumption that adjacent pixels in depth maps share a similar error pattern. First because adjacent pixels usually belong to same objects. Second, the error in LiDAR sensor has a correlation with distance from sensor, and as a result, pixels which are in an approximately equal distance to the sensor are likely to have close error magnitudes. From another perspective, the proposed slicing proved to be a simplistic segmentation based on the error pattern and was able to remarkably contribute to the correction stage.

According to Table 1, the results obtained when using L-BFGS as the optimizer are equal to ones without optimization on both KITTI and NYUv2 datasets. This means that L-BFGS could not improve the results because, unlike PSO, it does not have the capability for global search. In better words, it seems that it was trapped in local optima, i.e., the depth provided by SDC. Therefore, due to the identical performances and for the sake of conciseness, just one row is dedicated to both SDC and L-BFGS in Figure 7 and Figure 8.

In the meanwhile, PSO improved the results significantly in terms of all metrics and on both KITTI and NYUv2 datasets. For instance, PSO showed nearly 50% enhancement in AbsRel and 14% in RMSE on KITTI and 6% and 86%, respectively, in terms of AbsRel and RMSE on NYUv2.

If we compare the improvement of PSO over L-BFGS on KITTI and that on NYUv2, it can be observed that the improvement was more remarkable on NYUv2. Thus, considering that the baseline was trained on KITTI, one can conclude that the optimization module with PSO as its optimizer plays a significant role in the adaptation process. This observation also demonstrated the capability and efficacy of the activation optimization used in the proposed scheme.

To conclude, both of the proposed correction and optimization stages in DARS, i.e., SDC and activation optimization using PSO, proved to be effective and led to considerable improvements. Moreover, DARS proved its capability in network adaptation, given its performance on NYUv2.

As is clear from error patterns in Figure 7, related to KITTI and Figure 8 pertaining to NYUv2, the introduction of PSO has led to considerable improvements. The improvements can be specifically observed in more distant pixels which are usually of a higher error magnitude.

### 5.6. Comparison with SOTA

Proficient generalization is necessary for DL-based depth estimation methods, especially in applications with constantly changing environments, such as SLAM and autonomous vehicles. To deal with this problem, an inference-time refinement scheme is proposed to help pre-trained networks adapt to new inputs. To show the generalization performance of the proposed scheme, it has been compared with a range of unsupervised and supervised methods. On the other hand, to evaluate its adaptation performance, DARS with pre-trained weight on KITTI is applied on an unseen benchmark dataset, namely NYUv2. As is clear from Table 2, DARS outperformed competing methods in terms of almost all assessment criteria except for $\delta_{1.25^{2}}$ and $\delta_{1.25^{3}}$. From the perspective of these two criteria, the performance of our method was not as good as the second-place rival. However, DARS led to better performance in terms of $\delta_{1.25}$, which is the primary criterion for accuracy assessment. Although DARS utilizes a self-supervised baseline, Monodepth2, it outperformed its supervised rivals by a 39% margin in terms of RMSE on KITTI. This confirms the superiority of the proposed DARS even over supervised approaches and in dealing with harder scenes in a seen dataset. A visual comparison between DARS and the second best method in terms of RMSE on KITTI is presented in Figure 9.

Regarding the second dataset, NYUv2, DARS outperformed the competing methods in terms of all criteria according to Table 3. In terms of AbsRel and RMSE, DARS reached improvements of, respectively, 83% and 70% with respect to the best competing method. Furthermore, this table indicates how the proposed method successfully adapted to an unseen dataset. Note that unlike DARS, the other methods in Table 3 have been trained on NYUv2. Hence, one can deduce that DARS not only could adapt a network to an unseen dataset but also outperformed the methods trained on the exact same dataset. Furthermore, it suggests DARS as a possible alternative to supervised approaches which suffer from complicated generalization problems in practice. This adaptation capability is extremely advantageous in applications with constantly changing environments such as SLAM, where the scenes are of an unlimited variety and sparse LiDAR maps are available on the fly. A visual comparison between DARS and the second best method in terms of RMSE on NYUv2 is presented in Figure 10.

## 6. Conclusions

This paper deals with one of the main problems of available deep learning-based depth estimation networks, which is their limited generalization capability. This problem specifically restricts the practical usage of such models in applications with a constantly changing environment, such as SLAM. To alleviate this problem, a new double-stage adaptive refinement scheme for depth estimation networks, namely, DARS based on the combination of f-BRS and PSO, is proposed in this paper. DARS, here, is injected into Monodepth2 as the baseline and adapts the pre-trained network to each input during inference. Experimental results on KITTI and NYUv2 datasets demonstrated the efficacy of the proposed scheme not only for KITTI but also for NYUv2, while the baseline model was pre-trained only on KITTI. Although our approach is model agnostic by design, this paper did not explore the effects of using different baselines. In future work, we will, therefore, replace our unsupervised baseline with other networks, ranging from unsupervised to supervised in order to investigate the effectiveness of our proposed scheme on different baselines.

## Figures and Tables

**Figure 1 sensors-22-09755-f001:**
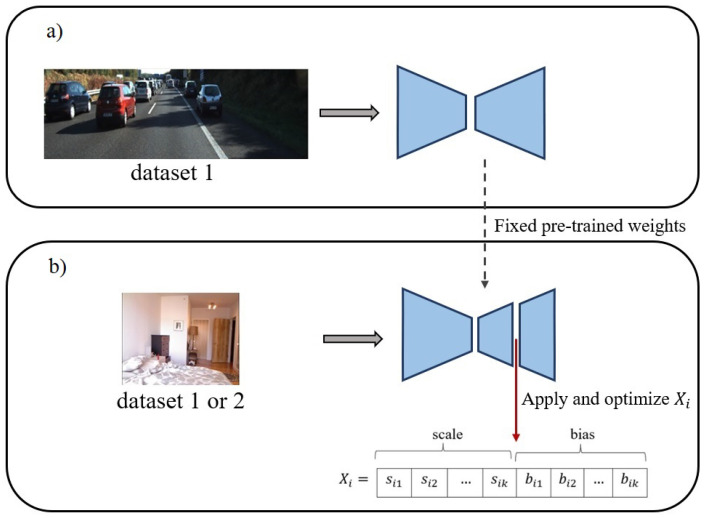
Overall performance of the proposed scheme: (**a**) Train on dataset 1, (**b**) inference on the same or another dataset. Keeping the pre-trained weights fixed, channel-wise scales and biases are applied to an intermediate activation map and are optimized to improve the predictions for the input dataset. The input dataset can be identical to the training dataset or a new one.

**Figure 2 sensors-22-09755-f002:**
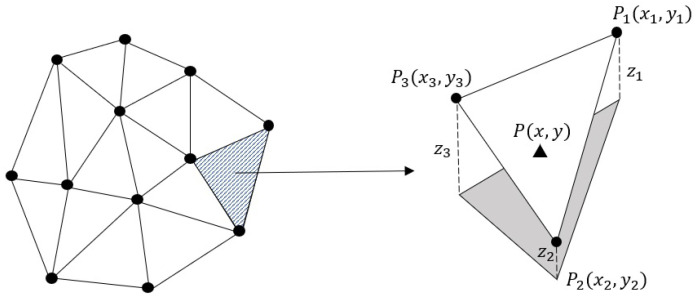
Delaunay-based interpolation on a set of points. First, a Delaunay triangulation is carried out on the points. Then, a plane is fitted to each triangle, and finally, the value for points on each of them is obtained based on the fitted plane.

**Figure 3 sensors-22-09755-f003:**
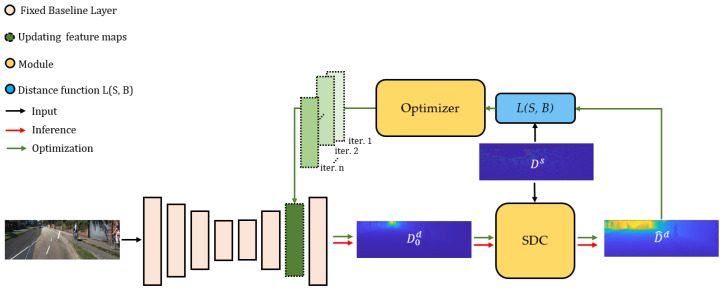
The double-stage adaptive refinement scheme (DARS). DARS refines depth maps estimated by baselines through optimizing (green arrows) an intermediate set of feature maps during inference (red arrows). Hence, given a pretrained baseline with fixed weights and an RGB input image, an initial depth $D_{0}^{d}$ is estimated. In the first stage (correction), a Sliced Delaunay Correction (SDC) module corrects $D_{0}^{d}$ with the guidance of a SD map $D^{s}$. Afterwards, an optimizer module (PSO) tries to update the intermediate feature maps to minimize the distance between the dense corrected depth ${\hat{D}}^{d}$ and $D^{s}$. As the scheme can be used for any deep architecture, the baseline (here, monodepth2) is illustrated minimally.

**Figure 4 sensors-22-09755-f004:**
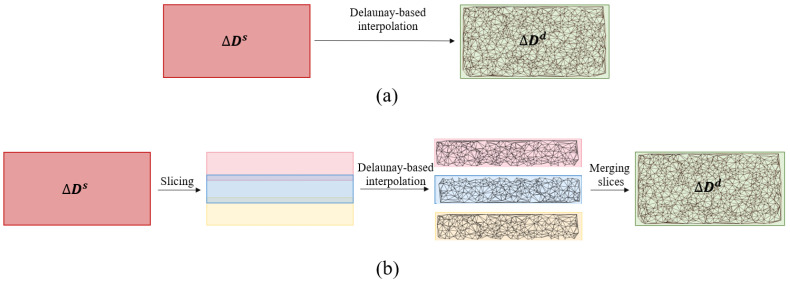
(**a**) Delaunay correction (DC), and (**b**) sliced Delaunay correction (SDC). In DC, the Delaunay-based interpolation is conducted on the whole sparse correction map. While in SDC, each SD map is first divided into three slices with overlap, then the correction value ($\Delta D^{s}$) is interpolated using Delaunay-based interpolation in each of them, independently. In the overlapped areas, the average of the values from the two slices is taken.

**Figure 5 sensors-22-09755-f005:**
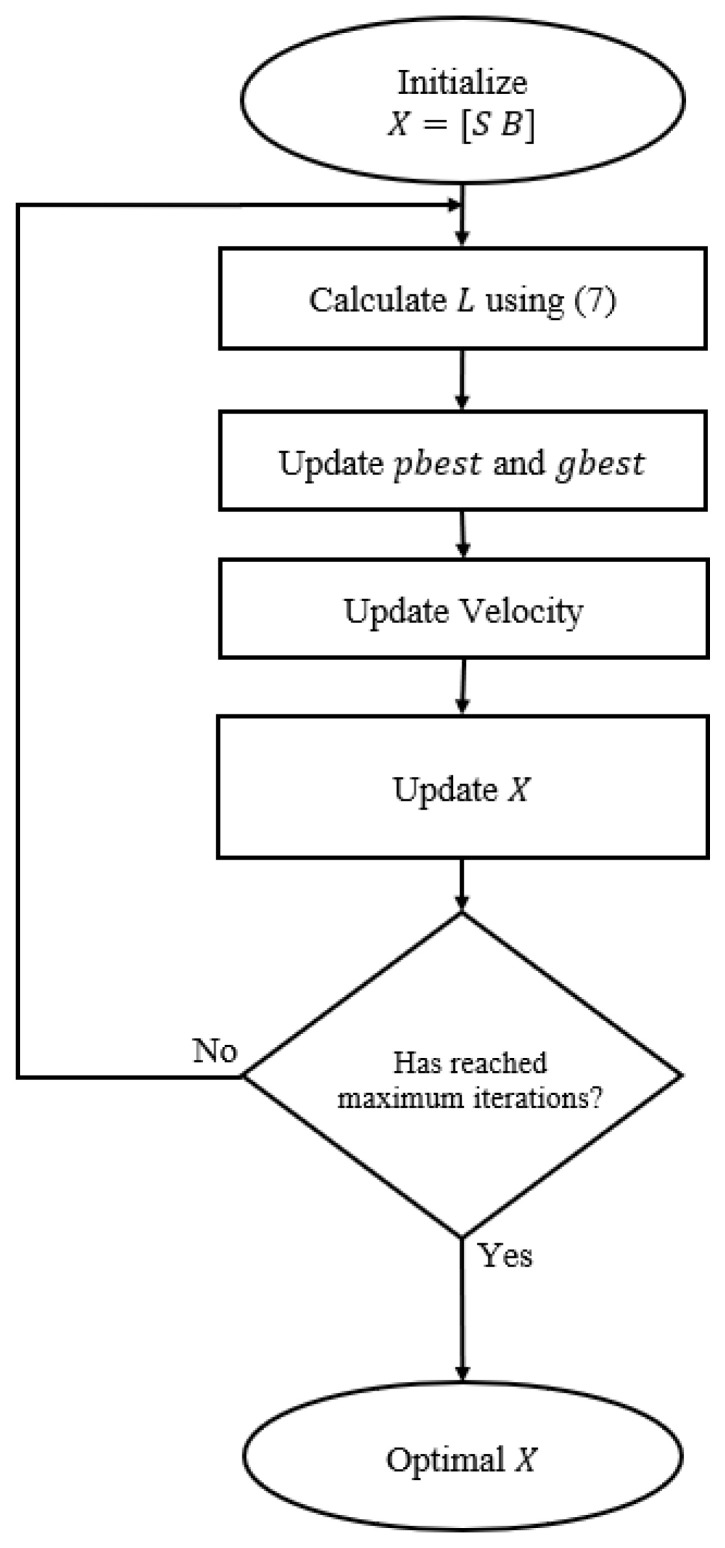
Flow chart of PSO in DARS, where position matrix *X* contains scales and biases. The theoretical background of PSO is provided in Section 3.2.

**Figure 6 sensors-22-09755-f006:**
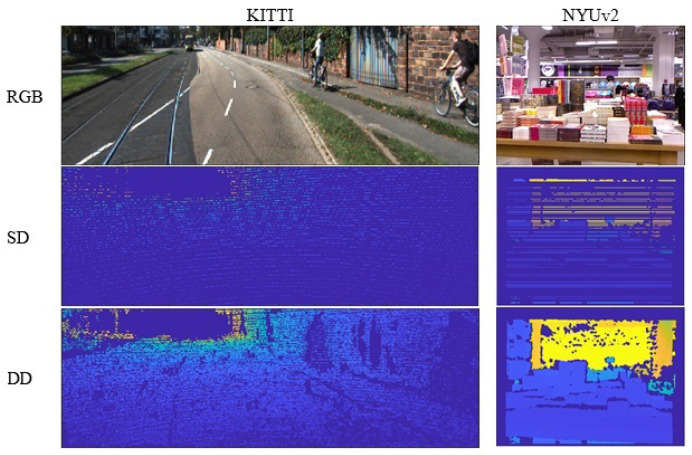
A sample of the used datasets.

**Figure 7 sensors-22-09755-f007:**
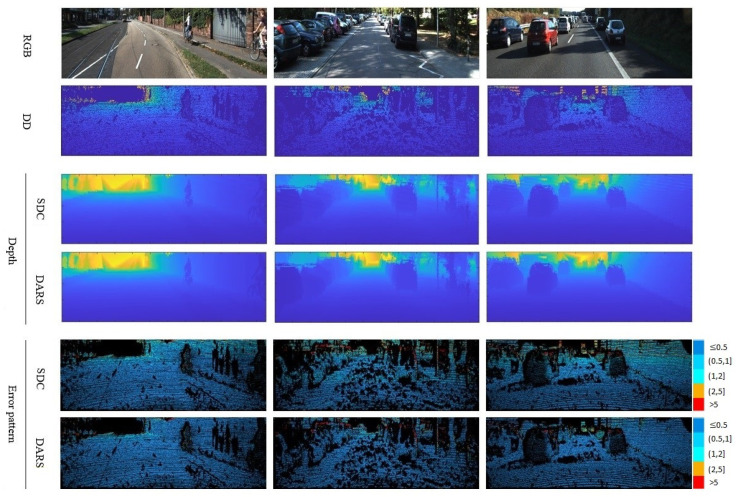
Visual results related to the ablation study of KITTI dataset. Numbers on the right side of error patterns are in meters.

**Figure 8 sensors-22-09755-f008:**
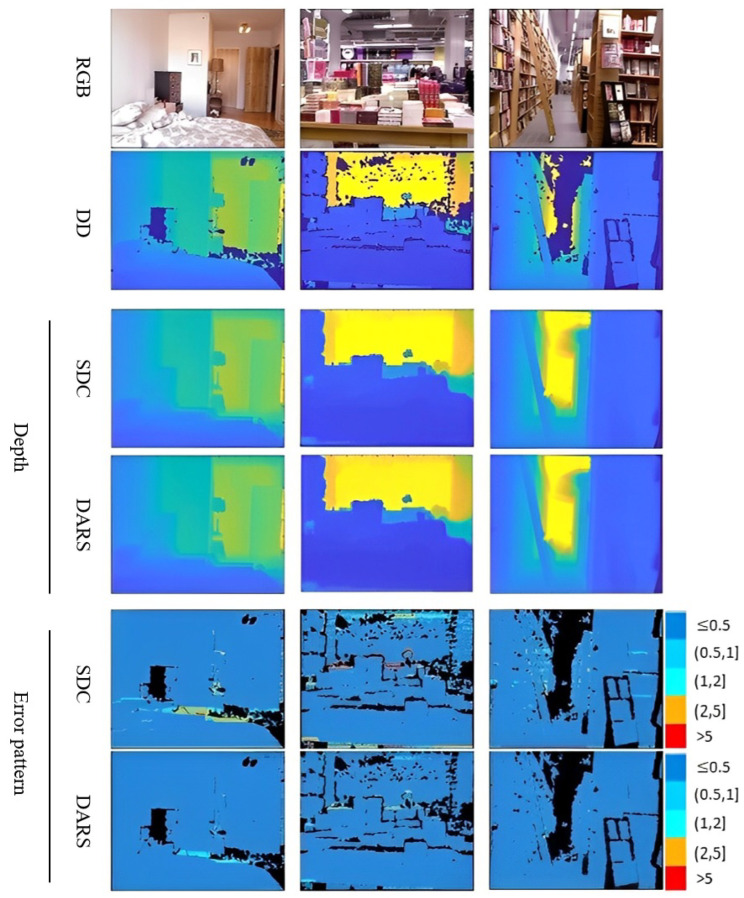
Visual results related to ablation study of NYUv2 dataset. Numbers on the right side of error patterns are in meters.

**Figure 9 sensors-22-09755-f009:**
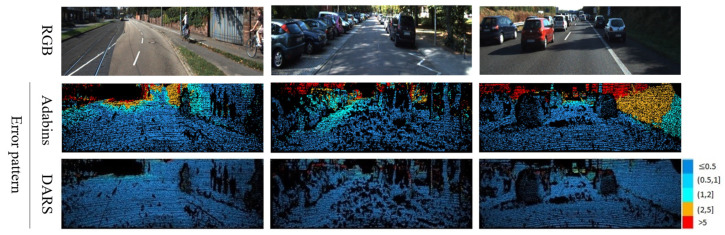
Visual results related to the comparative study of the KITTI dataset. The results of Adabins [14], as the second best method, are brought. Also, numbers on the right side of error patterns are in meters.

**Figure 10 sensors-22-09755-f010:**
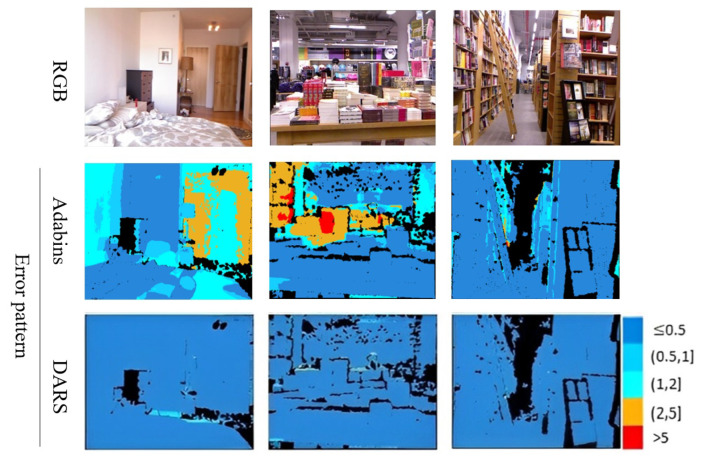
Visual results related to comparative study on NYUv2 dataset. The results of Adabins [14], as the second best method, are brought. Also, numbers on the right side of error patterns are in meters.

**Table 1 sensors-22-09755-t001:** Ablation study on KITTI and NYUv2. Monodepth2* is the version of Monodepth2 without median scaling.

Dataset	Modules			Lower Is Better			Higher Is Better		
	Baseline	Correction	Optimizer	AbsRel	RMSE	${\mathrm{RMSE}}_{\log}$	$\delta_{1.25}$	$\delta_{1.25^{2}}$	$\delta_{1.25^{3}}$
KITTI	Monodepth2	-	-	0.090	3.942	0.137	0.914	0.983	0.995
	Monodepth2*	-	-	0.996	19.324	5.715	0.000	0.000	0.000
	Monodepth2*	DC	-	0.864	16.888	3.149	0.183	0.330	0.447
	Monodepth2*	SDC	-	0.046	1.676	0.091	0.976	0.991	0.995
	Monodepth2*	SDC	L-BFGS	0.046	1.676	0.091	0.976	0.991	0.995
	Monodepth2*	SDC	PSO	**0.024**	**1.440**	**0.071**	**0.985**	**0.993**	**0.996**
NYUv2	Monodepth2*	SDC	-	0.018	0.766	0.747	0.972	0.974	0.975
	Monodepth2*	SDC	L-BFGS	0.018	0.766	0.747	0.972	0.974	0.975
	Monodepth2*	SDC	PSO	**0.017**	**0.109**	**0.044**	**0.993**	**0.996**	**0.999**

**Table 2 sensors-22-09755-t002:** Comparative study on KITTI. The first part from above contains unsupervised methods while the second part is dedicated to supervised ones.

Method	Lower Is Better				Higher Is Better		
	AbsRel	SqRel	RMSE	${\mathrm{RMSE}}_{\log}$	$\delta_{1.25}$	$\delta_{1.25^{2}}$	$\delta_{1.25^{3}}$
[49]	0.120	0.789	4.755	0.177	0.856	0.961	0.987
[28]	0.132	0.994	5.240	0.193	0.833	0.953	0.985
[26]	0.114	0.898	4.935	0.206	0.861	0.949	0.976
[23]	0.090	0.545	3.942	0.137	0.914	0.983	0.995
[50]	0.090	0.424	3.419	0.133	0.916	0.984	0.996
[51]	0.060	0.231	2.642	0.094	0.958	0.994	0.999
[14]	0.058	0.190	2.360	0.088	0.964	**0.995**	**0.999**
DARS	**0.024**	**0.137**	**1.442**	**0.071**	**0.985**	0.993	0.996

**Table 3 sensors-22-09755-t003:** Comparative study on NYUv2.The first part from above contains unsupervised methods while the second part is dedicated to supervised ones.

Method	Lower Is Better			Higher Is Better		
	AbsRel	RMSE	${\mathrm{RMSE}}_{\log}$	$\delta_{1.25}$	$\delta_{1.25^{2}}$	$\delta_{1.25^{3}}$
[47]	0.158	0.641	-	0.769	0.950	0.988
[15]	0.110	0.392	0.047	0.885	0.978	0.994
[51]	0.107	0.373	0.046	0.893	0.985	0.997
[14]	0.103	0.364	**0.044**	0.903	0.984	0.997
DARS	**0.017**	**0.109**	**0.044**	**0.993**	**0.996**	**0.999**

## Data Availability

Not applicable.

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
