# Peer review of "An Adaptive Refinement Scheme for Depth Estimation Networks"

_sensors, 2022, doi:10.3390/s22249755_

Round 1
Reviewer 1 Report
This manuscript proposes an inference-time adaptive refinement scheme for monocular depth estimation networks to improve their generalization capability. However, some modification and improvement need to be made in article organization and the experiment part. Specifically, the problems as follows.
(1) In Introduction, the order of Paragraphs 3 and 4 can be changed. Since the discussion object of Paragraph 3 is supervised learning-based methods rather than all deep learning methods.
(2) As described in Section 4, unsupervised architecture Monodepth2 is used during training, utilizing image triplet in a sequence and a pair of stereo images. And a sparse depth (SD) map in addition to an RGB image is used during inference. Since SD maps are not used for training but only for refinement in testing, this scheme cannot be classified as purely unsupervised or semi-supervised. As a result, the comparison of depth estimation results in Sections 5.5 and 5.6 is not conducted within the same category of methods. For example, [26] and [23] do not have SD maps in testing, [49] and [28] use other types of data (like optical flow or geometry constraint) for training. So, the comparison should be reconsidered.
(3) The method part (Section 4) is relatively short. The algorithm flow of PSO is not given.
(4) In Table 1 in Section 5.5, the gap between results of first two rows and next three rows on KITTI dataset is unreasonable. When conducting evaluation for depth estimation methods, we usually show the metrics after median scaling by ground truth depth maps. Without this, it seems that the resolution of scale ambiguity is contributed to SDC, which is not right. It is very likely that the remarkable contribution in metrics is largely due to the use of SD maps during inference.
(5) A detail needs to be modified: In Equation (12), accuracy should be the percentage of di s.t. δ < threshold.
Reviewer 2 Report
1. In the abstract section, please define “f-BRS” and rephrase the last sentence. Please avoid “outperform” and “rival methods”.
2. Keywords, in which sense optimization? Please add more details.
3. Please revise the second bullet in the list of contributions. The authors need to highlight the propose contribution. How is this concept novel?
4. Section 2 is missing several important citations in the depth estimation domain.
a. Wang, T.-C.; Efros, A.A.; Ramamoorthi, R. Depth estimation with occlusion modeling using light-field cameras. IEEE Trans. Pattern Anal. Mach. Intell. 2016, 38, 2170–2181.
b. Williem, W.; Park, I.K.; Lee, K.M. Robust Light Field Depth Estimation Using Occlusion-Noise Aware Data Costs. IEEE Trans. Pattern Anal. Mach. Intell. 2018, 40, 2484–2497.
c. Feng, M.; Wang, Y.; Liu, J. Benchmark data set and method for depth estimation from light field images. IEEE Trans. Image Process. 2018, 27, 3586–3598.
d. S. Heber and T. Pock, “Convolutional networks for shape from light field,” in Proc. IEEE Conf. Comput. Vis. Pattern Recognit., Jun. 2016, pp. 3746–3754.
e. Shin, C.; Jeon, H.; Yoon, Y.; Kweon, I.S.; Kim, S.J. EPINET: A Fully-Convolutional Neural Network Using Epipolar Geometry for Depth From Light Field Images. In Proceedings of the IEEE Conference on Computer Vision and Pattern Recognition, Salt Lake City, UT, USA, 18–23 June 2018; pp. 4748–4757
f. Schiopu, I.; Munteanu, A. Residual-error prediction based on deep learning for lossless image compression. IET Electron. Lett. 2018, 54, 1032–1034.
g. Ma, H.; Qian, Z.; Mu, T.; Shi, S. Fast and Accurate 3D Measurement Based on Light-Field Camera and Deep Learning. Sensors 2019, 19, 4399.
h. Rogge, S.; Schiopu, I.; Munteanu, A. Depth Estimation for Light-Field Images Using Stereo Matching and Convolutional Neural Networks. Sensors 2020, 20, 6188.
5. At the beginning of section 3, please add a paragraph describing how these concepts will be used in the proposed method.
6. Figure 3. Please add a few more details about the figure in the caption. Some of the blocks have no notation. I understand that these are well-known concepts, however, please add a few details as the figure remains a bit unclear for a non-expert.
7. Figure 4. Why three slices and not 2 or 4?
8. At the beginning of Section 4, please provide a more detailed overall description of the proposed method and how the building blocks are connected.
9. Figure 7 is a bit blurry.
10. A very detailed experimental section. The evaluation clearly proves that the proposed method provides an improved performance compared with state-of-the-art methods.
11. Would it be possible to have a visual comparison between proposed method and state-of-the-art methods? E.g. with [14].
12. In general, the manuscript is well-written and easy to follow.
Round 2
Reviewer 1 Report
The revised version of this manuscript reveal that the authors have made effort to polish their research work, and the quality of this version has been enhanced. Since the problems suggested to be addressed in the last review have been answered, I suggest to accept this version directly.